# B-Type Natriuretic Peptide—A Paradox in the Diagnosis of Acute Heart Failure with Preserved Ejection Fraction in Obese Patients

**DOI:** 10.3390/diagnostics14080808

**Published:** 2024-04-12

**Authors:** Marius Rus, Loredana Ioana Banszki, Felicia Liana Andronie-Cioara, Oana Liliana Pobirci, Veronica Huplea, Alina Stanca Osiceanu, Gheorghe Adrian Osiceanu, Simina Crisan, Decebal Dumitru Pobirci, Madalina Ioana Guler, Paula Marian

**Affiliations:** 1Department of Medical Disciplines, Faculty of Medicine and Pharmacy, University of Oradea, 410073 Oradea, Romania; rusmarius@uoradea.ro (M.R.); paula.marian85@gmail.com (P.M.); 2Faculty of Medicine and Pharmacy, University of Oradea, 410073 Oradea, Romania; banszkiloredana@gmail.com (L.I.B.); fcioara@uoradea.ro (F.L.A.-C.); opobirci@uoradea.ro (O.L.P.); hupleaveronica@uoradea.ro (V.H.); osiceanualina@yahoo.com (A.S.O.); osiceanuadrian@yahoo.com (G.A.O.); 3Department of Preclinical Disciplines, Faculty of Medicine and Pharmacy, University of Oradea, 410073 Oradea, Romania; 4Department of Psycho Neuroscience and Recovery, Faculty of Medicine and Pharmacy, University of Oradea, 410073 Oradea, Romania; 5Morphological Disciplines, Faculty of Medicine and Pharmacy, University of Oradea, 410073 Oradea, Romania; 6Cardiology Department, “Victor Babes” University of Medicine and Pharmacy, 2 Eftimie Murgu Sq., 300041 Timisoara, Romania; simina.crisan@umft.ro; 7Institute of Cardiovascular Diseases Timisoara, 13A Gheorghe Adam Street, 300310 Timisoara, Romania; 8Research Center of the Institute of Cardiovascular Diseases Timisoara, 13A Gheorghe Adam Street, 300310 Timisoara, Romania; 9The Municipal Hospital, “Dr. Pop Mircea” Marghita, 415300 Marghita, Romania; danpobirci@gmail.com

**Keywords:** B-type natriuretic peptide, obesity, congestive heart failure, New York Heart Association, dyspnea, fatigue

## Abstract

Background and objectives: B-type natriuretic peptide (BNP) represents a clinical tool for the diagnosis and prognostic evaluation of acute and chronic heart failure patients. The purpose of this retrospective study was to evaluate BNP values in obese and non-obese patients with acute heart failure with preserved ejection fraction. Materials and methods: In this study, we enrolled 240 patients who presented to the emergency department complaining of acute shortness of breath and fatigue. The patients were divided into two groups according to their body mass index (BMI) values. The BMI was calculated as weight (kilograms) divided by height (square meters). The BNP testing was carried out in the emergency department. Results: Group I included patients with a BMI of <30 kg/m^2^ and group II included patients with a BMI of ≥30 kg/m^2^. The average age of the patients was 60.05 ± 5.02 years. The patients in group II were significantly younger compared with those included in group I. Group II included a higher number of women compared to group I. Group I had fewer patients classified within New York Heart Association (NYHA) functional classes III and IV compared with group II. Echocardiography revealed an ejection fraction of ≥50% in all participants. Lower BNP levels were observed in patients from group II (median = 56, IQR = 53–67) in comparison to group I (median = 108.5, IQR = 106–112) (*p* < 0.001). Conclusions: Obesity and heart failure are continuously rising worldwide. In this retrospective study, we have highlighted the necessity to lower the threshold of BNP levels in obese patients with acute heart failure and preserved ejection fraction.

## 1. Introduction

An increased likelihood of heart failure with preserved ejection fraction (HfpEF) has been associated with obesity, which is also a risk factor for hypertension and coronary artery disease [1,2,3]. Body fat has a major impact on the diagnostic and prognostic values of multiple parameters. It has been suggested that obese patients with heart failure have lower natriuretic peptide levels due to the increased expression of clearance receptors and augmented peptide degradation by the adipose tissue [4].

Diagnosing HFpEF remains difficult. The diagnosis of HFpEF is defined as a left ventricular ejection fraction of ≥50%, along with objective evidence of cardiac structural and/or functional abnormalities, consistent with the presence of left ventricle (LV) diastolic dysfunction/raised LV filling pressures [5,6,7]. Heart failure (HF) has become the most common cardiovascular condition in hospitalized patients, while the number of HF patients with significant obesity has increased dramatically. This may be associated with the elevated prevalence of obesity in the general population. For this reason, a more precise understanding of the inter-relationship between body mass index and B-type natriuretic peptide (BNP) levels is very important for a more accurate diagnosis of heart failure in obese patients [8].

In developed countries, the age-adjusted incidence of heart failure may be decreasing, presumably reflecting the better management of cardiovascular disease, despite the increasing overall incidence [9,10,11,12]. Currently, the incidence of HF in Europe is about 3/1000 person-years (all age-groups) or about 5/1000 person-years in adults [13,14]. More than 50% of patients diagnosed with HF are female [15]. The circulating biomarkers, BNP and N-terminal-pro B-type natriuretic peptide (NT-pro BNP), are used to confirm or exclude the diagnosis of heart failure in patients with acute dyspnea in the emergency department. Cardiac etiology of dyspnea—more specifically, HF—is strongly associated with higher natriuretic peptide levels [16].

As it has been suggested that obese patients may have lower natriuretic peptide concentrations, potentially affecting the diagnosis of HF, the aim of this study was to assess BNP levels in obese and non-obese patients with acute heart failure with preserved ejection fraction.

## 2. Materials and Methods

### 2.1. Study Design and Study Approval

A total of 240 patients with heart failure were included in this retrospective study. The participants were admitted to the Cardiology Clinic of the Bihor County Emergency Hospital between January 2020 and January 2023.

The data were obtained with the approval of the ethics committee of the Bihor County Clinical Emergency Hospital (No. 82/16 June 2019).

### 2.2. Enrollment of Patients and Data Collection

The body mass index (BMI) was calculated for all patients (weight in kilograms divided by height in meters squared). Overweight, or pre-obesity, was defined as a BMI of 25–29.9 kg/m^2^, while a BMI of ≥30 kg/m^2^ equaled obesity. The clinical criteria for the diagnosis of heart failure were as follows: dyspnea, reduced exercise tolerance, hepatomegaly, jugular venous distension, lower extremity edema, ascites, and pulmonary congestion, which were determined based on the patient’s medical history and physical examination. The paraclinical tests included lung X-rays, standardized 2D echocardiography, and laboratory tests (BNP serum levels). The left ventricular ejection fraction was echocardiographically calculated using the Simpson method. Blood samples were collected and placed into ethylenediaminetetraacetic acid (EDTA) tubes and processed immediately. The B-type natriuretic peptide (BNP) values were measured using enzyme-linked immunosorbent assay (ELISA) technology, and the name of the analyzer was Compact Immuno-Analyzer Pathfast. The detection limit was 4 pg/mL. All patients had this analysis performed on admission. The BNP level was the last parameter determined for each patient, in order not to influence the diagnosis of heart failure. In this study, we selected heart failure patients with an ejection fraction of 50% or more. The diagnosis of heart failure with preserved ejection fraction (HfpEF) was established based on the following criteria: the presence of signs and/or symptoms of heart failure, the chest X-ray data (pulmonary congestion, interstitial edema, alveolar edema, and pleural effusion), the estimated risk of HFpEF (all patients enrolled in this study presented high probabilities of HfpEF), the presence of diastolic dysfunction of the left ventricle, reflected as elevated filling pressure E/e’ of >15 mmHg, and BNP values testing.

The exclusion criteria were as follows: underweight patients (BMI less than 18.5 kg/m^2^), admission for acute right heart failure secondary to pulmonary embolism (BNP values being increased in these cases), and a left ventricle ejection fraction of <50%. In this study, we enrolled 240 patients, divided into 2 groups. Group I consisted of patients with a BMI of under 30 kg/m^2^ and group II included patients with a BMI higher than 30 kg/m^2^.

### 2.3. Statistical Analysis

All of the data from this study were analyzed using IBM SPSS Statistics 25 and illustrated using Microsoft Office Excel/Word 2021. The quantitative variables were tested for normal distribution using the Shapiro–Wilk test and were written as averages with standard deviations or medians with interquartile ranges. The quantitative independent variables with non-parametric distribution were tested between groups using Mann–Whitney U tests, and correlations between them were quantified using Spearman’s rho correlation coefficients. The qualitative variables were written as counts or percentages, and the differences between groups were tested using Fisher’s Exact test. Z-tests with Bonferroni corrections were used to further detail the results obtained in the contingency tables. The measure of associations was quantified as odds ratios with 95% confidence intervals when applicable. Univariable and multivariable linear regression models were used for predicting the BNP. The models were tested for validity and significance, while the performance of the prediction was written as beta coefficients with 95% confidence intervals for each variable.

## 3. Results

### 3.1. Patients Characteristics

The average age of the patients was 60.05 ± 5.02 years. In group I, the average age was 63.05 ± 2.74 years, and in group II, the average age was 57.04 ± 5 years. As can be seen, the patients with obesity from group II were younger than those with a BMI of <30 kg/m^2^. This retrospective study included 240 patients, in which 140 (58.33%) were women and 100 (41.67%) were men.

Group I included 120 patients—63 (52.5%) women and 57 (47.5%) men, with 60 patients presenting a BMI of <25 kg/m^2^ and the remaining 60 displaying a BMI of between 25 kg/m^2^ and 30 kg/m^2^, along with symptoms of congestive heart failure. Echocardiographic assessments revealed an LV ejection fraction (EF) of 51–60% in 70 patients and over 60% in 50 patients. The value of BNP was <90 pg/mL in 7 patients, between 90 and 100 pg/mL in 11 patients, and over 100 pg/mL in 102 patients.

Group II included 120 patients—77 (64.2%) women and 43 (35.8%) men. The entire group had a BMI of ≥30 kg/m^2^, with symptoms of congestive heart failure. Echocardiographic assessments revealed an LV EF of 51–60% in 80 patients and >60% in 40 patients. The value of BNP was <50 pg/mL in 8 patients, between 50 and 70 pg/mL in 89 patients, and between 70 and 110 pg/mL in 23 patients.

The female patients displayed a slightly increased association with the higher BMI group (≥30 kg/m^2^) (64.2% vs. 35.8%), although the differences between genders were not statistically significant (*p* = 0.089) (Table 1).

The differences between the groups were statistically significant according to the Fisher’s Exact Test (*p* < 0.001), and Z-tests with Bonferroni correction showed that the patients with a BMI index lower than 30 kg/m^2^ were significantly more associated with New York Heart Association (NYHA) II class than NYHA III or NYHA IV classes (25% vs. 45.8%/29.2%), while the patients with a BMI index higher than or equal to 30 kg/m^2^ were significantly more associated with NYHA III or NYHA IV classes than the NYHA II class (66.7%/33.3% vs. 0%). (Table 1).

### 3.2. Clinical and Paraclinical Correlations

The data in Table 2 show the comparison of the BNP values between the other factors analyzed in this study. The results show that the BMI group, gender, existence of hypertension, and NYHA classes have a significant influence on BNP values. Age is also significantly correlated with BNP (*p* < 0.001, R = 0.555—Spearman’s rho correlation coefficient), showing a significant positive moderate power correlation.

From the point of view of the objective examination of group I, 37 (30.83%, *p* = 0.02) patients had diastolic gallop, 45 (37.5%, *p* = 0.0358) had hepatomegaly with liver pain, 93 (77.5%, *p* < 0.001) had leg edema, 23 (19.17%, *p* = 0.027) had jugular turgor, 17 (14.17%, *p* = 0.07) had ascites, and 114 (95%, *p* = 0.28) had pulmonary stasis.

In group II, 61 (50.83%) patients had diastolic gallop, 53 (44.17%) had hepatomegaly (assessed by ultrasound), 116 (96.67%) had leg edema, 39 (32.5%) had jugular turgor, 29 (24.17%) had ascites (also assessed by ultrasound), and 118 (98.33%) had pulmonary stasis. (Figure 1).

From an imaging point of view, the radiological descriptions were classified according to the Ross criteria. In group I, 37 patients (30.83%) fit into the first grade of the Ross classification (circulatory flow redistribution), while 45 patients (37.5%) pertained to the second grade (loss of structure of vascular and Kerley B lines) and 21 patients (17.5%) to the third grade (localized pulmonary edema). The remaining 17 patients (14.17%) were part of the fourth grade (diffuse pulmonary edema). In group II, 19 patients (15.83%) were classified as grade I, 35 patients (29.17%) as grade II, 40 patients (33.33%) as grade III, and 26 patients (21.17%) as grade IV.

The data presented in Figure 2 show the distribution of the patients according to the BMI index and Ross criteria. The differences between the groups were statistically significant according to the Fisher’s Exact Test (*p* = 0.002), and the Z-tests with Bonferroni correction showed that the patients with a BMI index lower than 30 kg/m^2^ were significantly more associated with Ross grade I than grade III or grade IV (66.1% vs. 37.7%/34.9%), while the patients with a BMI index higher than or equal to 30 kg/m^2^ were significantly more associated with Ross grade III or grade IV than grade I (62.3%/65.1% vs. 33.9%).

Regarding the echocardiographic examinations, 70 patients (58.30) in group I presented an LV EF of 51–60%, while, in group II, 80 patients (66.70%) displayed the same EF value. An LV EF of over 60% was discovered in 50 patients (41.70%) in group I and 40 patients (33.30%) in group II. The differences between the groups were not statistically significant (*p* = 0.230—Fisher’s Exact Test), as shown in Figure 3.

The distribution of the BNP parameter was non-parametric in both BMI groups, according to the Shapiro–Wilk test (*p* < 0.001). The differences between the groups were statistically significant, according to the Mann–Whitney U Test (*p* < 0.001), in which the patients with a BMI index higher than or equal to 30 kg/m^2^ had a significantly lower BNP (median = 56, IQR = 53–67) than the patients with a BMI index lower than 30 kg/m^2^.

The lower BNP values obtained in group II were in total contradiction with the diagnosis of these patients, with most of them being diagnosed with congestive heart failure functional classes New York Heart Association (NYHA) III and IV (Table 3).

Linear regression models were used for predicting BNP, with the multivariable model showing that the male patients (β = 8.27, 95% C.I.: 4.81–11.72) and patients with a BMI below 30 kg/m^2^ (β = 43.94, 95% C.I.: 39.54–48.35) are at risk for increased BNP, while adjusting for other variables (Table 4).

## 4. Discussion

The prevalence of heart failure (HF) and obesity has increased over the last decades. For an accurate diagnosis and management of HF, it is important to consider the relationship between body mass index (BMI) and B-type natriuretic peptide (BNP) values. The present study was prompted by earlier observations in other studies, suggesting that, in patients with congestive heart failure (CHF) and a BMI of ≥30 kg/m^2^, the BNP values present a lower sensitivity as a diagnosis tool for CHF, at least regarding a value of 100 pg/mL.

The aim of this study was to assess the BNP levels in obese and non-obese patients with acute heart failure with preserved ejection fraction. This retrospective study was conducted on 240 patients (140 women and 100 men) presenting with acute dyspnea and fatigue. Four major clinical presentations can be described, according to the 2021 European Society of Cardiology (ESC) guidelines. Acute decompensated heart failure (ADHF) is the most common form of acute heart failure, accounting for 50–70% of presentations [17,18,19]. Acute pulmonary edema is related to lung congestion. The clinical criteria for the diagnosis of acute pulmonary edema include the following: dyspnea and orthopnea, respiratory failure (hypoxemia/hypercapnia), tachypnea (>25 breaths/min), and increased respiratory effort [20]. Right ventricle (RV) failure is associated with increased RV and atrial pressures, associated with systemic congestion. RV failure may also impair left ventricle filling and ultimately reduce systemic cardiac output through ventricular interdependence [21]. The definition of cardiogenic shock presents this syndrome as a result of a primary cardiac dysfunction, following an inadequate cardiac output and creating a life-threatening state of tissue hypoperfusion, which can eventually lead to multi-organ failure and death [22,23,24].

It is already known that obesity is a major risk factor for the development of heart failure, regardless of gender [25]. Obese individuals present lower plasma natriuretic peptide levels compared to individuals with a normal BMI. This cannot be explained by the differences in cardiovascular risk factors or cardiac structure between obese and non-obese patients. This observation is supported by its consistency across natriuretic peptides, BNP and N-terminal-pro B-type natriuretic peptide (NT pro-BNP), in both genders (male and female), as well as the separate analyses focused on low natriuretic peptide values. In women, abdominal obesity further predicts lower natriuretic peptide levels, even after BMI adjustment is applied [26,27].

Obese patients present an expanded intravascular volume, associated with an increased cardiopulmonary volume. As a result, an elevated BNP level is to be expected. However, obese patients with heart failure seem to have lower BNP levels, perhaps in relation to the non-hemodynamic factors. The exact pathophysiological relationship between low BNP and obesity is still unknown. Numerous mechanisms are associated with the endocrine release of cytokines, or “adipokines”, by adipocytes [28]. Gentili et al. showed in their study that the adipose cells of obese patients had less natriuretic peptide receptor-A (NPR-A) and a higher concentration of natriuretic peptide receptor-C (NPR-C) [29]. Therefore, there would be a greater likelihood of decreased cellular effects and an increased clearance of circulating BNP in obese patients. They demonstrated a negative correlation between this low NPR-A/NPR-C ratio and insulinemia, insulin resistance, and BMI. Additionally, adipose tissue releases higher levels of proinflammatory interleukin 6 (IL-6), and the cells in adipose tissue exposed to IL-6 express higher levels of NPR-C and nearly half that of NPR-A [29]. One of the main factors for the disbalance in NPRs in obese people may be IL-6 secretion. Adipose tissue secretes pro-inflammatory cytokines, including resistin, interleukin-1β (IL-1β), and tumor necrosis factor-alpha (TNF-α), in addition to IL-6. These cytokines accelerate the breakdown of BNP and exacerbate cardiac fibrosis and atheromatosis [30].

Obesity is an important and independent factor influencing peripheral BNP expression in CHF patients [28]. As mentioned before, BNP and NT pro-BNP have been included in the diagnosis criteria of heart failure with preserved ejection fraction in the latest ESC guidelines [31]. The rationale behind this was to exclude several potential causes for dyspnea in patients with preserved left ventricular ejection fraction.

However, recent evidence indicates that BNP may be low in obese HF patients with preserved ejection fraction, which may represent a distinct phenotype of heart failure with preserved ejection fraction [32].

In our study, we have presented an increased incidence of HF patients, functional New York Heart Association (NYHA) class III and IV, pertaining to group II. The BNP levels fell between 50 and 70 pg/mL in group II for the majority of patients. In group I, the majority of BNP values raised above 100 pg/mL, confirming the accuracy of the measurements.

The lower BNP values obtained in group II totally contradicted the condition of these patients, with most of them being diagnosed with congestive heart failure functional class NYHA III and IV. The diagnosis was established independently of the B-type natriuretic peptide values. Based on this observation, it can be inferred that, in obese patients, the cut-off values for the BNP levels used for the diagnosis of heart failure should be lowered to between 50 and 70 pg/mL. Recent studies suggest that BNP levels are decreased in obese patients, due to changes in clearance receptors and accelerated peptide degradation [33]. The Suita study of the urban Japanese population involved 1759 subjects without atrial fibrillation or a history of ischemic heart disease, aged 38–95 years [34]. This study demonstrated a negative relationship between BNP values and obesity-related markers, such as body fat mass, skinfold thickness, and waist circumference. The results from the Breathing Not Properly Multinational Study suggested a lower cut-point level of BNP, more specifically, a minimum of 54 pg/mL, for the diagnosis of acute heart failure in severely obese patients [35]. A higher cut-point of B-type natriuretic peptide of 170 pg/mL or more, in lean patients, was suggested in order to increase specificity. The Breathing Not Properly Multinational Study was a prospective study of 1586 patients from seven centers who presented to the emergency room with acute dyspnea. Another earlier work, including 318 patients with heart failure, by Mehra MR showed similar results [36]. According to the results, the BNP levels were significantly lower in obese patients compared to non-obese patients (205 ± 22 and 335 ± 39 pg/mL, respectively; *p* = 0.0007).

The limitations of our study include the small number of participants, as well as the absence of specific relevant data that were not collected during the presentation to the emergency department.

## 5. Conclusions

Heart failure (HF) is the most common cardiovascular diagnosis among hospitalized patients. B-type natriuretic peptide values are frequently used to confirm the diagnosis of heart failure. Globally, the prevalence of HF in obese patients is continuously increasing. The conclusions of this study indicate that a lower cut-off of BNP levels should be used in the diagnosis of heart failure with preserved ejection fraction in obese patients. Otherwise, the risk of misdiagnosis remains high.

## Figures and Tables

**Figure 1 diagnostics-14-00808-f001:**
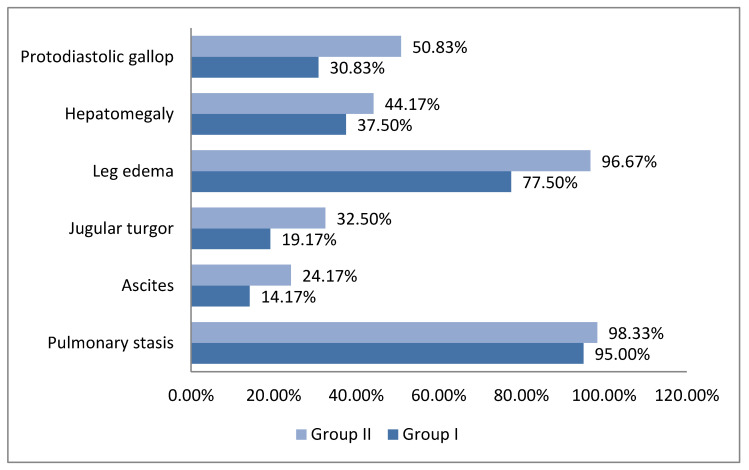
Distribution of patients according to symptoms of heart failure.

**Figure 2 diagnostics-14-00808-f002:**
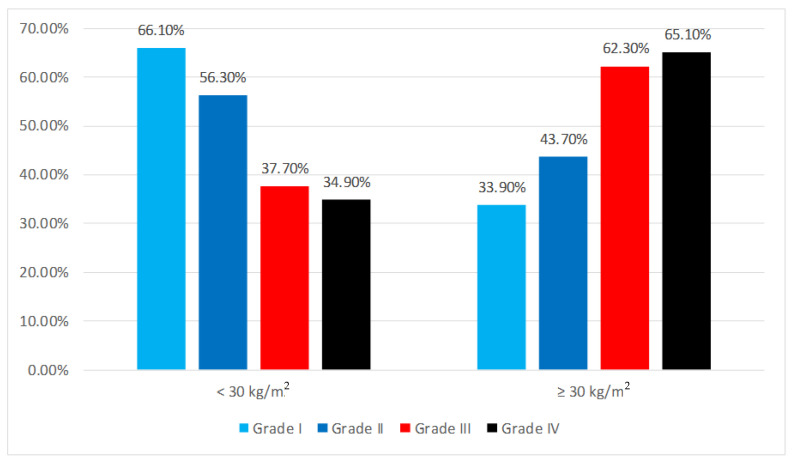
Distribution of patients according to BMI index and Ross criteria.

**Figure 3 diagnostics-14-00808-f003:**
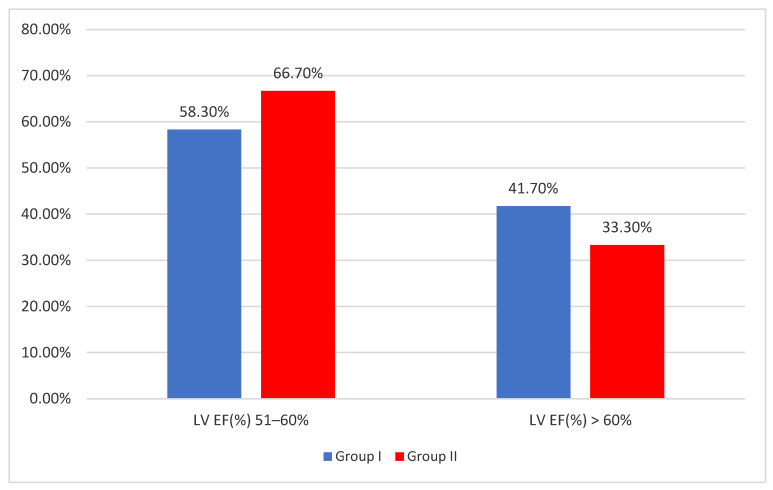
Distribution according to BMI index groups and LV EF (%) values.

**Table 1 diagnostics-14-00808-t001:** Distribution of the patients according to gender, BMI index, and NYHA classification.

Gender/BMI	Female		Male				*p*
	Nr.	Percent %	Nr.		Percent %		
<30 kg/m^2^	63	52.5	57		47.5		0.089
≥30 kg/m^2^	77	64.2	43		35.8		
BMI/NYHA class	NYHA II		NYHA III		NYHA IV		*p*
	Nr.	Percent %	Nr.	Percent %	Nr.	Percent %	
<30 kg/m^2^	30	25	55	45.8	35	29.2	<0.001
≥30 kg/m^2^	0	0	80	66.7	40	33.3	

**Table 2 diagnostics-14-00808-t002:** Comparison of BNP values between the other factors analyzed in this study.

BMI Group/BNP	Average ± SD	Median (IQR)	Mean Rank	*p* *
<30 kg/m^2^	107.12 ± 11.34	108.5 (106–112)	177.92	<0.001
≥30 kg/m^2^	62.09 ± 15.1	56 (53–67)	63.08	
Gender/BNP	Average ± SD	Median (IQR)	Mean Rank	*p* *
Female	78.85 ± 26.1	67 (55–108)	107.49	0.001
Male	92.66 ± 24.25	105 (75–110)	138.72	
Hypertension/BNP	Average ± SD	Median (IQR)	Mean Rank	*p* *
Absent	80.47 ± 26.12	69.5 (55–107)	109.93	0.020
Present	88.6 ± 25.72	105 (58–109)	130.73	
Diabetes/BNP	Average ± SD	Median (IQR)	Mean Rank	*p* *
Absent	85.14 ± 26.3	95 (56–109)	122.52	0.291
Present	81.65 ± 25.76	84 (55–106.5)	109.42	
CKD/BNP	Average ± SD	Median (IQR)	Mean Rank	*p* *
Absent	83.68 ± 26.11	87 (56–109)	120.47	0.994
Present	85.61 ± 26.37	98 (55–108)	120.53	
Rhythm status/BNP	Average ± SD	Median (IQR)	Mean Rank	*p* *
Sinus rhythm	81.53 ± 26.63	76 (54–108)	112.37	0.0504
Atrial fibrillation	88.18 ± 25.34	102 (58–110)	129.95	
NYHA/BNP	Average ± SD	Median (IQR)	Mean Rank	*p* **
Class II	105.07 ± 14.74	109 (98–112)	171.58	<0.001 ***
Class III	82.13 ± 25.9	84 (55–108)	114.19	
Class IV	80.88 ± 26.87	75 (55–108)	111.43	

* Mann–Whitney U Test, ** Kruskal–Wallis H Test, CKD = chronic kidney disease, *** Post hoc Dunn–Bonferroni tests show significant differences between Class II and Class III (*p* < 0.001) or Class II and Class IV (*p* < 0.001), differences between Class III and Class IV are not significant (*p* = 1.000).

**Table 3 diagnostics-14-00808-t003:** Comparison of BNP according to BMI index and NYHA classes in patients with a BMI index higher than or equal to 30 kg/m^2^.

BMI index	Average ± SD	Median (IQR)	Average Rank	*p* *
<30 kg/m^2^	107.12 ± 11.34	108.5 (106–112)	177.92	<0.001
≥30 kg/m^2^	62.09 ± 15.1	56 (53–67)	63.08	
NYHA class	Average ± SD	Median (IQR)	Average Rank	*p* *
NYHA III	64.7 ± 16.81	56 (53.2–69.5)	64.71	0.061
NYHA IV	56.88 ± 9	55 (51–63)	52.09	

* Mann–Whitney U Test.

**Table 4 diagnostics-14-00808-t004:** Univariable and multivariable linear regression for predicting BNP.

Variable	Univariable		Multivariable	
	β (95% C.I.)	*p*	β (95% C.I.)	*p*
BMI group (<30 kg/m^2^)	45.02 (41.63–48.42)	<0.001	43.94 (39.54–48.35)	<0.001
Gender (Male)	13.81 (7.27–20.35)	<0.001	8.27 (4.81–11.72)	<0.001
Hypertension	8.12 (1.52–14.72)	0.016	−0.05 (−3.51–3.41)	0.977
NYHA Class				
Class II (Reference)	-	-	-	-
Class III	−22.94 (−32.93–−12.94)	<0.001	3.58 (−1.87–9)	0.197
Class IV	−24.18 (−34.88–−13.49)	<0.001	−0.18 (−5.9–5.54)	0.951
Age	2.83 (2.27–3.39)	<0.001	0.14 (−0.26–0.54)	0.484

## Data Availability

The data presented in this study are available upon request from the corresponding authors.

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
