# Peer review of "B-Type Natriuretic Peptide—A Paradox in the Diagnosis of Acute Heart Failure with Preserved Ejection Fraction in Obese Patients"

_diagnostics, 2024, doi:10.3390/diagnostics14080808_

Round 1

Reviewer 1 Report (Previous Reviewer 1)

Comments and Suggestions for Authors

The authors have addressed the comments.

Author Response

Thank you very much

Reviewer 2 Report (New Reviewer)

Comments and Suggestions for Authors

    The introduction is too long and  descriptive of pathophysiology. It  would be better to move the part  underlined starting  from _  

  THE EXACT PATHOPHYSIOL ....    

to  discussion part

 How  was  diagnosed HfpEF'  it is not indicated, by HF  score , or filling pressure algorhythm

NT proBNP    and BNp are used   simultaneously as martkers   which makes   somewhat  confusion  , taking into consideration that the  diagnostic  challenge  already  exists in obese patients  based on this parameter .

 Conclusions  are  not based on  presented results and not supported by data .

Comments on the Quality of English Language

 quality of English is not good

Author Response

Reviewer 3 Report (New Reviewer)

Comments and Suggestions for Authors

Dear authors, I find the subject of your study intriguing; however, there are some suggested revisions that could enhance the quality of the article. Specifically, I noticed that the number of tables and figures is quite high, resulting in repeated duplications of information across the text, tables, and figures within the results section. It might be beneficial to consolidate certain tables into a single table and remove any duplicated results to streamline the presentation of your findings.

Round 2

Reviewer 2 Report (New Reviewer)

Comments and Suggestions for Authors

  Extensive revision has been provided

 Both  abstract  and introduction ,  results  have been reviewed according to comments, remarks

Comments on the Quality of English Language

Quality of  Englsih is  sufficient

This manuscript is a resubmission of an earlier submission. The following is a list of the peer review reports and author responses from that submission.

Round 1

Reviewer 1 Report

Comments and Suggestions for Authors

The manuscript explores the B-Type Natriuretic Peptide (BNP) paradox in the diagnosis of heart failure with preserved ejection fraction (HFpEF) in patients with obesity. The authors investigated this in a group of overweight and obese patients with heart failure and report differences in the distribution of the subjects based on obesity groups, sex, NYHA functional class, ROSS criteria and also the differences in ejection fraction and BNP levels between the groups. The results are as expected and support the paradoxical behaviour of BNP in obese subjects. The results highlight the need for adjusting BNP threshold cut-offs to accurately diagnose HFpEF in individuals with obesity.

As age and obesity independently influence BNP levels, a combined consideration is required to provide a more accurate reflection of the individual’s cardiovascular health status. The statistical analyses need to be adjusted for sex, age, and other covariates that could influence BNP values.

Additionally, it would have been more informative to include results on diabetes, hypertension, and renal function since these comorbidities should also be considered into diagnostic considerations.

Other minor points below.

Introduction: Percentage of men and women in group I and II is incorrect.

Section 3.1: reported mean age and standard deviation of the patients (all, group 1, group 2) seem to be incorrect.

The manuscript needs to be reread for several typos and grammatical errors.

Comments on the Quality of English Language

The manuscript needs to be reread for several typos and grammatical errors.

Reviewer 2 Report

Comments and Suggestions for Authors

The Authors decided to analyze the BNP values relationship with BMI in patients with HFpEF. The impact of obesity on cardiovascular diseases and outcomes is a very important area of research. However, the Authors mention larger studies with the same conclusions and similar results. Please explain what is the novelty of the presented study compared to previous studies in this area.

Moreover, the authors underline in the introduction section, that different tissue proportions can impact HF outcomes. BMI is a parameter that does not differ from obesity and edema resulting from heart failure. Do the authors assess the patients with any other, more specific tool, with enabled this adjustment?

The presentation or the results, background, and rationale of this study explanation is not clear enough, please provide the relevant information. Moreover, many statements lack relevant citations.

Please add line numbers to easier navigate within the manuscript.

Below you can find my comments, questions and suggestions.

Abstract

“Group II included a higher number of women compared to group I.” – please provide RR with p-value for this statement.

“Lower BNP levels were observed in patients from group II” – please provide relevant statistical analysis with p value.

Please provide the results (numbers) in the abstract, not only conclusions.

Introduction

In the introduction, please provide the background regarding BNP levels pathophysiology in HF and the potential impact of fat tissue on its metabolism. That would explain the rationale of this study.

Please provide the background on the HF and HFpEF diagnosis according to ESC.

“The exclusion criteria were: underweight patients (BMI less than 18,5 kg/m2), “ – why was this group excluded?

2.1. Was written consent taken from the participants?

“These BMI thresholds were proposed by a World Health Organization (WHO) expert report (WHO/NUT/NCD, 2000). “ - - please provide relevant citation.

“The study included 240 patients which were distributed into 2 groups: group I included the patients with BMI<30 kg/m2 ( non-obese patients) and group II patients with BMI 30 kg/m2 ( obese patients).” – were these groups matched? If yes, what was the matching parameter?

Please explain what figure 3 is supposed to present – mean, median EF? I suggest using boxplots in this case.

Why did the authors used BNP instead of NT-pro-BNP?

“Recent data suggest an inverse relationship between BNP levels and BMI values. „ – please provide relevant citations. If there is data in this area, what is novel in the presented study?

“Obese individuals have lower plasma natriuretic peptide levels compared with individuals with a normal BMI, which could not be explained by differences in cardiovascular risk factors or cardiac structure between obese and non-obese patients. This observation is sustained by its consistency across both natriuretic peptides (BNP and NT proBNP), in both gender (male and female), and also, by separate analyses focusing on low natriuretic peptide values „ – provide citations

“As it was mentioned, B-type natriuretic peptide and N-terminal-pro B-type natriuretic peptide have been included in the diagnosis crite- ria of heart failure with preserved ejection fraction in the latest European Society of Cardiology guidelines. “ – provide details of the cut-off values, and citation.

In your study, was BNP value adjusted to sinus ryths/atrial fibrillation status?

“Based on this observation, and considering similar observations by the American

Heart Association (AHA), „ – provide citation.

Then, “lowering cut-off values to 70” – the already cut-off value is 35.

Please correct the formatting.

Comments on the Quality of English Language

Minor corrections needed
